# Integrated Biomarker Analysis Reveals L1CAM as a Potential Stratification Marker for No Specific Molecular Profile High-Risk Endometrial Carcinoma

**DOI:** 10.3390/cancers14215429

**Published:** 2022-11-03

**Authors:** Antonella Ravaggi, Davide Capoferri, Laura Ardighieri, Iacopo Ghini, Federico Ferrari, Chiara Romani, Mattia Bugatti, Laura Zanotti, Stephanie Vrede, Germana Tognon, Johanna M. A. Pijnenborg, Enrico Sartori, Stefano Calza, Eliana Bignotti, Franco Odicino

**Affiliations:** 1Department of Clinical and Experimental Sciences, University of Brescia, 25123 Brescia, Italy; 2Angelo Nocivelli’ Institute of Molecular Medicine, ASST Spedali Civili di Brescia, University of Brescia, 25121 Brescia, Italy; 3Division of Obstetrics and Gynecology, ASST Spedali Civili di Brescia, 25123 Brescia, Italy; 4Department of Pathology, ASST Spedali Civili di Brescia, 25123 Brescia, Italy; 5Department of Medical and Surgical Specialties, Radiological Sciences and Public Health, University of Brescia, 25123 Brescia, Italy; 6Department of Obstetrics and Gynecology, Radboudumc, 6525 GA Nijmegen, The Netherlands; 7Unit of Biostatistics and Bioinformatics, Department of Molecular and Translational Medicine, University of Brescia, 25123 Brescia, Italy

**Keywords:** high-risk endometrial carcinoma, molecular classification, NSMP, L1CAM, prognosis, platinum-based adjuvant chemotherapy

## Abstract

**Simple Summary:**

Here, we showed the independent prognostic value of the four molecular subgroups—POLE-mutated, MMR-deficient, p53-abnormal, ‘no specific molecular profile’ (NSMP)—on a cohort of high-risk endometrial cancer patients. L1 neuronal cell adhesion molecule (L1CAM) expression could further stratify the NSMP subgroup, with L1CAM-positive patients having the worst prognosis compared to all other molecular subgroups. All NSMP/L1CAM-positive patients were “early-relapsing”, showing a significantly shorter platinum-free interval than L1CAM-negative patients after adjuvant platinum-based chemotherapy. Since the NSMP is the most heterogeneous subgroup, we believe that L1CAM may represent a relevant candidate biomarker to complement both prognostic stratification and prediction of chemotherapy benefit in patients with high-risk endometrial cancer.

**Abstract:**

Histopathologic assessment of high-risk endometrial cancer (EC) suffers from intersubject variability and poor reproducibility. The pragmatic classification in four molecular subgroups helps to overcome these limits, showing a significant prognostic value. The “no specific molecular profile” (NSMP) is the most heterogeneous EC subgroup, requiring further characterization to better guide its clinical management. DNA sequencing of POLE exonuclease domain and immunohistochemistry for PMS2, MSH6, and p53 were performed in order to stratify a cohort of 94 high-risk EC patients in the four molecular subgroups. Moreover, a panel of seven additional biomarkers was tested. Patients were found to be 16% POLE-mutated, 36% mismatch repair-deficient, 27% p53-abnormal, and 21% NSMP. In the multivariable model, molecular groups confirmed their significant association with disease-specific survival and progression-free survival, with p53-abnormal and NSMP endometrial cancer characterized by poor outcomes. Among the additional evaluated biomarkers, L1CAM was the only one with a significant prognostic value within the NSMP subgroup. NSMP/L1CAM-positive patients experienced the worst outcome and were “early-relapsing” after platinum-based chemotherapy, with a significantly shorter platinum-free interval compared to L1CAM-negative patients. L1CAM appears to be a promising candidate as a prognostic and predictive biomarker in the high-risk NSMP subgroup, which is actually known to lack specific molecular markers.

## 1. Introduction

Endometrial cancer (EC) is the most commonly diagnosed gynecologic neoplasm in Western countries, and its incidence is progressively rising due to increased age and obesity in the population [1,2]. Though most EC patients have an early-stage disease and a favorable prognosis, about 15–20% present with high-risk tumors characterized by non-endometrioid histology, a high-grade, advanced FIGO stage and increased cancer-related recurrence and mortality. Adjuvant treatment following surgical resection and staging is recommended for this aggressive EC subgroup, even if the identification of patients who would benefit from therapeutic options among radiotherapy, chemotherapy or a combination of both remains an unanswered question [3]. Unfortunately, histomorphological assessment of high-risk ECs suffers intersubject variability and poor reproducibility within and between cancer centers, leading to difficulties in the identification of these patients [4]. To improve the current risk stratification tools, a molecular classification of EC in four subgroups, showing differences both in genomic profile and in progression-free survival, was recently introduced by the Cancer Genome Atlas (TCGA) [5]. Subsequently, pragmatic assays were developed, significantly overlapping and correlating with the outcomes found in the four groups defined by the TCGA [6,7]. The four prognostic subgroups include: DNA polymerase epsilon (POLE) exonuclease domain-mutated, mismatch repair-deficient (MMR-D), p53-abnormal (p53abn) and no specific molecular profile (NSMP) ECs. Tumors with POLE mutations have the most favorable prognosis, those with p53abn display the worst outcome, while NSMP and MMR-D are characterized by intermediate survival probabilities. The molecular classification has been applied also to high-risk ECs, where the presence of the four molecular subgroups was confirmed [8,9,10]. Among those, NSMP is the most heterogeneous group and the one for which novel biomarkers of risk stratification are urgently needed. To this aim, in the present study we have applied the aforementioned molecular classification to a cohort of high-risk ECs and further investigated the expression of additional markers with known prognostic potential in EC—estrogen (ER) and progesterone (PR) receptors [11,12], L1 Cell Adhesion Molecule (L1CAM) [12,13,14,15], Catenin Beta 1 (CTNNB1) [16], AT-Rich Interaction Domain 1A (ARID1A) [17,18], ki-67 proliferation index [19], and intratumoral immune cell infiltrate [20]—within the four molecular subgroups. Specifically, we analyzed the potential of those additional markers to further stratify the NSMP category, characterized by marked heterogeneous clinicopathologic features and prognosis.

## 2. Materials and Methods

### 2.1. Patient and Samples

A retrospective study was performed on a cohort of 94 EC patients (Brescia Cohort) diagnosed and treated at the Division of Obstetrics and Gynecology of the ASST Spedali Civili of Brescia (Italy). The inclusion criteria were as follows: histologically confirmed EC, classified as high–intermediate, high or advanced-metastatic risk classes [21]; and availability of endometrial tumor tissues. All the eligible patients treated at our institution between January 2004 and October 2019 were included in the study. All patients were followed from the time of their confirmed diagnosis until death, or August 2021. Clinical and histopathological data were acquired from the original reports and outpatient appointments. A validation cohort of 47 EC patients (ENITEC Cohort) who met the inclusion criteria was obtained by the European Network for Individualized Treatment of Endometrial Cancer (ENITEC) consortium. Patients’ characteristics are summarized in Appendix B, Table A1.

### 2.2. Immunohistochemistry

All cases were reviewed by two pathologists (L.A. and I.G.) and classified according to the latest WHO guidelines. For each case, the best representative slide of the tumor was chosen, and the corresponding block was then retrieved for immunohistochemical (IHC) analysis. IHC was performed on formalin-fixed paraffin-embedded (FFPE) tumor sections, with antibodies specific to PMS2, MSH6, p53, ER, PR, Ki67, ARID1A, and L1CAM (see Section B.1 for detailed procedures and scoring). Briefly, abnormal expression of one or both PMS2 and MSH6 proteins was considered as indicating MMR-D, whereas p53 strong positive nuclear staining of more than 80% of tumor cells, or complete loss of expression, were considered to indicate aberrant/mutation-type (p53abn). L1CAM membranous staining in more than 10% of tumor cells was set as the threshold for L1CAM positivity, following an established scoring system [22]. The nuclear expression of ARID1A, was considered intact (wild type) if present in the entire neoplastic population and lost (mutated) if there was a lack of expression both widespread and clonal. The assessment of immune cell infiltrate was performed in 10 high-power fields (40X) with a semiquantitative method evaluating the overall cell density of mononuclear cells, including lymphocytes and plasma cells, in the stromal peritumoral areas. Immune cell infiltrate was graded as: absent, weak, moderate and prominent.

### 2.3. DNA Extraction and Next-Generation Sequencing

Genomic DNA was extracted from 94 FFPE tissues, each containing at least 70% of tumor cells, starting from one 10 µm-thick section. DNA extraction was performed using the GeneRead DNA FFPE kit, according to manufacturer’s instructions (Qiagen, Hilden, Germany), and DNA samples were quantified with a Qubit fluorometer (ThermoFisher Scientific, Waltham, MA, USA).

Next-generation sequencing (NGS) was used to assess the mutational status of POLE (exons 9, 10, 11, 12, 13, 14, 19, 20, 26, 30, 36, 37, 39, 42, 46) and CTNNB1 (exon 3) with a custom AmpliSeq gene panel (Appendix A) on Ion 316 v2 chips (Thermo Fisher Scientific). Data analysis was performed using Ion Reporter v5.18.0.2 software (Thermo Fisher Scientific). Only patients with pathogenic variants in the POLE exonuclease domain have been included in the POLE mutant subgroup [23]. For more details on NGS analyses, see Section B.2. Quality control data of NGS output files for each patient, as well as cumulatively, in terms of total coverage, on-target sequencing, average depth and uniformity are shown in Appendix A.

### 2.4. Statistics

Quantitative data were described using the mean, standard deviation (SD), median, and interquartile range (IQR). Qualitative variables were summarized using absolute counts and percentages. The association between the clinicopathological/molecular characteristics and the molecular subgroups was assessed with Pearson’s Chi-squared test with simulated *p* values (B = 2000 Monte Carlo simulations) and a Kruskal–Wallis rank-sum test for categorical and quantitative variables, respectively. Progression-free survival (PFS) was taken to be the time elapsed from surgery to progression, recurrence or death, whereas disease-specific survival (DSS) was defined as the time from surgery to cancer-related death or the last follow-up.

Platinum-free interval (PFI) was calculated from the end of platinum-based adjuvant chemotherapy to the date of recurrence/progression. For all endpoints, the last date of follow-up was used for censored subjects. Patients were dichotomized into “late-relapsing” and “early-relapsing” categories based on whether they had a PFI >12 months or <6 months, respectively. PFS and DSS analysis were performed using univariable and multivariable Cox regression models, while the Kaplan–Meier estimator was used for survival curve calculation and display.

Model validation on external data was performed after the model update using logistic calibration [24]. Briefly, we assumed that the baseline survival might be different and that some model coefficients might be either over- or underestimated, even though they still have the same relative effect. Concordance between observed and estimated survival in the validation cohort was evaluated using the extension of the C-index for the censored outcome [25].

All statistical tests were two-sided, and a 5% significance level was assumed. All analyses were performed using R (version 4.2.1) [26]. 

## 3. Results

### 3.1. Molecular Classification of High-Risk EC Patients Is Associated with Clinicopathological Characteristics

The assignment of EC to the four molecular subgroups was performed according to the most recent literature [9,27], and was successful in all 94 subjects. The list of POLE pathogenic variants and all other nonpathogenic variants is displayed in Appendix A, respectively.

Table 1 displays the distribution of tumor specimens into the four molecular subclasses: 16% POLE-mutated, 36% MMR-D, 27% p53abn and 21% NSMP. Six out of 94 (6%) patients showed more than one molecular feature and were classified according to Leon-Castillo et al. [27]: 4 subjects were both POLE-mutated and MMR-D and were classified as POLE-mutated, 1 subject was POLE-mutated and p53abn and was classified as POLE-mutated, and 1 MMR-D and p53abn subject was classified as MMR-D (Appendix A).

Tumor histotype, grade, FIGO stage, lymph node status, and the ESGO/ESTRO/ESP 2020 risk group exhibited a significant association with the molecular subgroups (Table 1). In detail, EC patients with a pathogenetic variant in the POLE exonuclease region were predominantly characterized by early-stage, high-grade, endometrioid histotypes, as well as the absence of lymph node metastasis. The MMR-D subgroup mainly showed endometrioid histotype, a high grade, and a homogeneous distribution among early and advanced stages. The p53abn tumors were significantly enriched in non-endometrioid histotype, high-grade, and advanced stages. NSMP ECs were more frequently endometrioid, with advanced stage and with the highest rate of lymph node involvement and myometrial invasion >50%. We evaluated the association between molecular stratification and the time to recurrence in the subset of patients who received platinum-based adjuvant chemotherapy after surgery, classified as “late-relapsing” or “early-relapsing” based on their PFI. POLE-mutated and MMR-D ECs revealed a greater proportion of “late-relapsing” patients (86% and 70%, respectively), while NSMP and p53abn subgroups showed a homogeneous distribution among “late-relapsing” and “early-relapsing” patients (47% vs. 53% and 45% vs. 55%, respectively, *p* = 0.14). We observed a similar trend using PFI as the endpoint. Indeed, considering MMR-D as the reference group, only the p53abn group showed a significantly shorter time for recurrence after chemotherapy (HR = 2.28, 95%CI: 1.03–5.01, *p* = 0.041), while NSMP patients and the POLE-mutated group did not (HR = 2.07, 95%CI: 0.89–4.82, *p* = 0.091 and HR = 0.21, 95%CI = 0.03–1.65, *p* = 0.14, respectively).

### 3.2. Molecular Classification Is an Independent Prognostic Factor in High-Risk ECs

All 94 patients were considered for survival analysis (median follow up time, 81.3 months, range 9–209 months). Fifty patients (53.2%) had disease recurrence or progression. At the time of the last follow up, 51 patients (54.3%) were alive, 36 (38.3%) succumbed to the disease and 7 (7.4%) died from other causes. In univariable survival analysis, all clinicopathological parameters known to be associated with worse prognosis were significantly correlated with shorter DSS and PFS: older age, non-endometrioid histological type, presence of lymph node metastasis, advanced FIGO stage, and a higher ESGO/ESTRO/ESP 2020 prognostic risk group (Appendix B, Table A2). As expected, molecular groups differed significantly both in terms of DSS and PFS, with POLE-mutated and MMR-D tumors characterized by more favorable outcomes compared to p53abn and NSMP subgroups (Table 2 and Appendix D Figure A1). Of note, “multiple-classifier” [27] patients had an outcome similar to that of the molecular subgroup where we reclassified them. Indeed, only one out of the four POLE mut/MMR-D patients experienced relapse, the one POLE mut/p53abn never relapsed, and all of them showed an excellent clinical outcome, as did the MMR-D/p53abn patient (Appendix A).

Multivariable analysis included age, stage and grade as clinical characteristics. As the vast majority of patients were LVSI positive (85 out of 94), it was not included in the multivariate model. Molecular groups confirmed their significant association with DSS and PFS (Table 2). Of note, the exception was represented by the NSMP group, which showed a substantial reduction in the hazard ratio (HR) when predicting DSS in multivariable analysis (HR = 1.98, *p* = 0.142, Table 2).

### 3.3. Additional Biomarkers Are Significantly Associated with the Molecular Classification

We next investigated the expression of the additional biomarkers ER, PR, Ki-67, CTNNB1, ARID1A, L1CAM, and immune infiltrate. As shown in Table 3, the percentage of ER and PR positive cells was significantly different among the four groups (*p* = 0.023 and 0.002, respectively), being lower in p53abn and higher in MMR-D tumors. Greater expression of ER and PR was associated with endometrioid histology (*p* = 0.002 and *p* < 0.001), L1CAM negative expression (both *p* < 0.001), and late time to recurrence after platinum-based adjuvant chemotherapy (*p* = 0.018 and *p* = 0.002). Furthermore, higher ER positivity was found in patients without lymph node metastases (*p* = 0.035), while higher PR positivity correlated with CTNNB1 mutated EC (*p* = 0.037) (Appendix B, Table A3 and Table A4).

The percentage of Ki-67 positive cells demonstrated a significant association with the molecular subgroups, with higher value in POLE-mutated ECs and lower in NSMP tumors (*p* = 0.012, Table 3). Furthermore, a higher cell proliferation index correlated with grade 3 ECs on the whole cohort (*p* = 0.022, Appendix B, Table A3).

Mutations in CTNNB1 exon 3 were found exclusively in MMR-D (18%) and NSMP (30%) subgroups, while all POLE-mutated and p53abn tumors were characterized by wild-type CTNNB1 (*p* = 0.003, Table 3). CTNNB1 mutational status correlated with histology (*p* = 0.032) and p53 (*p* = 0.017), as all variants were present in endometrioid and p53wt ECs (Appendix B, Table A4). The complete list of CTNNB1 variants is displayed in the Appendix A and most of them (11 out of 14 variants) have already been described in EC [28,29].

Loss of ARID1A expression was observed mainly in the POLE, MMR-D and NSMP groups and only in a minority of p53abn patients, (*p* = 0.008, Table 3). Moreover, ARID1A-loss significantly correlated with younger age (*p* = 0.004), p53wt (*p* = 0.016) and L1CAM-negative expression (*p* = 0.035, Appendix B, Table A4).

The proportion of L1CAM-positive subjects was higher in p53abn, followed by NSMP, POLE-mutated and MMR-D tumors (*p* < 0.001, Table 3). The main clinicopathological and molecular parameters that correlated with L1CAM positivity were non-endometrioid histological type (*p* < 0.001), advanced FIGO stage (*p* = 0.041), G3 tumor grade (*p* = 0.018), high and advanced risk group (*p* = 0.002), p53abn (*p* < 0.001), MMR proficiency (*p* < 0.001), lower expression of ER and PR (both *p* < 0.001), and presence of ARID1A expression (*p* = 0.035, Appendix B, Table A4).

The density of intratumoral immune cell infiltrate was significantly different within the molecular subgroups, with greater presence of moderate/prominent inflammation in POLE-mutated ECs and less in NSMP and p53abn tumors (*p* = 0.001, Table 3). Furthermore, higher levels of immune cell infiltrate correlated with early stage (*p* = 0.007) and late time to recurrence after platinum-based adjuvant chemotherapy (*p* = 0.002). Absent/weak inflammation was observed predominantly in tumors belonging to high and advanced metastatic risk groups of patients (*p* = 0.012, Appendix B, Table A4).

### 3.4. L1CAM Is a Predictor of Worse Prognosis in the Whole EC Patients’ Cohort and in the NSMP Subgroup

The additional biomarkers described in the above paragraph were considered for univariable survival analysis both in the whole EC patients’ cohort and within each molecular subgroup. In the whole cohort, L1CAM positivity was significantly associated with a shorter DSS (HR = 2.70, *p* = 0.003), whereas a higher percentage of ER (HR = 0.98, *p* = 0.015) and PR (HR = 0.97, *p* = 0.006) positive cells, loss of ARID1A (HR = 0.43, *p* = 0.036) and the presence of abundant intratumoral inflammatory infiltrate (HR = 0.43, *p* = 0.018) were found to be significant protective factors for DSS. Considering PFS, the presence of a moderate/prominent immune cell infiltrate was confirmed as a protective factor (HR = 0.34, *p* = 0.001). L1CAM positivity (HR = 1.67, *p* = 0.072) and ARID1A loss (HR = 0.58, *p* = 0.091) achieved marginal significance in predicting shorter and longer PFS, respectively (Table 4).

The univariable survival analysis within each molecular class showed that, among all additional biomarkers, PR positivity showed a protective effect for PFS in the NSMP subgroup (HR = 0.84, *p* = 0.045). Within the same molecular class, L1CAM was found to be a significant risk factor for both DSS (HR = 3.93, *p* = 0.045) and PFS (HR = 4.12, *p* = 0.023, Appendix A). Accordingly, we stratified the NSMP subgroup based on the L1CAM status and we found that the NSMP/L1CAM-positive patients were characterized by the worst outcome (*p* = 0.009 and *p* = 0.001 for DSS and PFS, respectively), even poorer than p53-abn group (Table 5, Figure 1). All NSMP/L1CAM-positive patients were characterized by advanced stage, high grade and positive lymph nodes. Notably, 80% of them showed ER and PR negative staining and ARID1A expression. By contrast, 27% of NSMP/L1CAM-negative patients were early stage, 47% G2 and 36% with negative lymph nodes. Furthermore, 73% of them were characterized by ER and PR positivity and ARID1A loss in 47% of cases. In multivariable analysis, NSMP/L1CAM-positive status was confirmed to be an independent negative prognostic factor for PFS (*p* = 0.024) and, marginally, for DSS (*p* = 0.056). As expected, POLE mut and p53abn status were positive and negative independent prognostic factors, respectively, in terms of both DSS and PFS (Table 5).

We validated the survival model for PFS on an independent cohort (ENITEC Cohort) of 47 high-risk EC patients, harboring 1 POLE, 7 MMR-D, 22 NSMP, and 17 p53abn tumors. We can observe a substantially different survival rate in the validation cohort, especially in the NSMP subgroup, where only 18% of patients relapsed or died of the disease. Due to this different survival rate, we recalibrated the survival models developed on the Brescia cohort using logistic calibration. The predicted 36 months PFS rates using the model accounting for L1CAM were 0.63 and 0.66 for the NMSP L1CAM-positive and L1CAM-negative patients, respectively, with an observed PFS rate of 0.75 and 0.90. The model not accounting for L1CAM (based only on molecular classification) reported a predicted survival of 0.66 for L1CAM-positive and 0.64 for L1CAM-negative groups. While the estimates still underestimate the true survival rate, due to the substantial difference between the two cohorts, adding L1CAM has actually achieved a better ranking within the NSMP subgroups. This suggests a lower PFS, something coherent with the observed results, for the L1CAM-positive group. Moreover, the concordance indexes (C-indexes) between observed and predicted survival for the NSMP subgroup for the two models were 0.75 for the extended model with L1CAM, versus 0.68 for the baseline model. Thus, our results indicate that L1CAM expression, when added to the baseline model, improves the performance in predicting recurrence.

### 3.5. L1CAM Correlates with the Time to Recurrence after Platinum-Based Chemotherapy in NSMP High-Risk ECs

We further investigated whether the stratification of the NSMP subgroup based on L1CAM status, besides being a prognostic factor, could also be informative of the time to recurrence after the end of platinum-based adjuvant chemotherapy. In the NSMP subgroup, L1CAM-negative patients were found to be “late-relapsing” after platinum-based chemotherapy in 70% of cases with a median PFI of 37 months (range 14–69). In contrast, all L1CAM-positive patients were “early-relapsing”. The difference in the time until recurrence after platinum-based chemotherapy between these two groups was significant (Chi-square Test, *p* = 0.026), and L1CAM-positive patients showed a significantly shorter PFI than L1CAM-negative ones (Figure 2, HR = 4.00, 95%CI = 1.20–13.3, *p* = 0.025).

## 4. Discussion

The introduction of the pragmatic molecular classification in EC arose from the need to overcome the reproducibility limits of current clinicopathological parameters in the risk stratification of patients [4,6,7]. Recently, the prognostic value of such molecular classification has been reported also in high-risk ECs in independent cohorts and clinical trials [7,8,9,10]. The results of the present study confirm the independent prognostic value of the molecular classification in patients with high-risk ECs. Indeed, POLE-mutated patients were characterized by an excellent prognosis, even when a high tumor grade or an advanced stage of disease were present. The clinical course of POLE-mutated ECs has been recently attributed to its intrinsic indolent biology, which confers to the patients an optimal outcome even if classified as high-risk and without any adjuvant treatment [30]. These findings support the rationale of two ongoing randomized clinical trials, PORTEC-4a [31] and RAINBO (ClinicalTrials.gov Identifier: NCT05255653), investigating adjuvant radiotherapy and chemotherapy de-escalation in women with POLE-mutated ECs. In our cohort, women with p53abn tumors showed poor clinical outcomes, while subjects belonging to the MMR-D subgroup had intermediate survival rates, results which were consistent with the literature [8,9,10]. The NSMP ECs, that were traditionally classified in the intermediate prognosis group [6,7], showed a particularly unfavorable outcome, similar to p53abn patients. This is likely related to the patients selected in our study, that include mostly high-grade, advanced-stage, biologically aggressive tumors, and is in line with previous findings by Leon-Castillo et al. [30]. The NSMP EC category is characterized by the absence of specific molecular markers, being p53 and POLE wild-type, as well as MMR-proficient. Besides, it is a heterogeneous group, likely composed by a mixture of distinct subgroups, characterized by different risk of recurrence and potential susceptibility to tailored treatments.

In the search for additional biomarkers useful for further NSMP prognostic stratification [12,13,14,15,16,17,18,19,20,32], we found that L1CAM expression was significantly associated with a poor outcome, regardless of tumor grade and FIGO stage. More specifically, NSMP/L1CAM-positive patients showed an extremely poor prognosis and features typical of aggressive tumors, such as the absence of ER/PR expression and ARID1A positivity. By contrast, the NSMP/L1CAM-negative patients were mainly ER/PR-positive and ARID1A-negative, and showed an intermediate prognosis, likely attributable to a more favorable underlying tumor biology. Our results are in agreement with previous studies, which showed, in cohorts including mostly low-risk tumors, L1CAM expression as an independent poor prognostic factor in EC patients [14,15] and confirmed its potential in risk classification of the NSMP subgroup. [33]. Our data suggest that L1CAM expression might be considered an independent adverse prognostic factor also in high-risk NSMP ECs.

Our findings point to the potential involvement of L1CAM in the platinum chemotherapy response in NSMP ECs treated in the adjuvant setting, since NSMP/L1CAM-positive patients were all characterized by an “early-relapsing” disease and by a significantly shorter PFI than L1CAM-negative ones. Consistent with this, our group recently reported L1CAM overexpression as an independent predictor of poor response to platinum-based chemotherapy in two independent cohorts of high-risk EC patients, suggesting the incorporation of L1CAM evaluation into the treatment decision-process for the identification of women who were not benefiting from platinum treatment [34]. Thus, L1CAM expression could be informative of the time to recurrence after platinum-based adjuvant chemotherapy also in the molecularly heterogeneous NSMP group.

Our findings on the potential prognostic and predictive role of L1CAM expression in high-risk NSMP ECs, if confirmed in wider patients’ cohorts, could have important clinical implications. The evaluation of L1CAM expression in NSMP ECs via biopsy tissues taken at the moment of diagnostic hysteroscopy might help in identifying patients benefiting from a more extended surgical approach due to the aggressive biology of NSMP/L1CAM-positive tumors, which is comparable or even worse than that of p53abn ECs. Due to this elevated risk of recurrence, patients with high-risk NSMP tumors might be eligible for adjuvant chemotherapy after surgery. In the treatment decision process, the evaluation of L1CAM expression along with clinical and molecular information might help in identifying high-risk NSMP EC patients who are potentially sensitive to platinum chemotherapy and women who, instead, might benefit from alternative treatments. For instance, NSMP/L1CAM-negative patients were also characterized by higher ER and PR expression, as well as ARID1A loss, all of which could be therapeutic targets for hormonal treatment and/or immunotherapy [35,36,37]. On the contrary, human monoclonal antibody (MAb) approaches might be investigated in NSMP/L1CAM-positive patients, as already reported in various types of cancer [38,39,40]. In those patients, characterized by an “early-relapsing” disease after platinum-based treatment, the combination of L1CAM MAbs with chemotherapy might be explored, following the rationale of tumor cell resensitization to standard chemotherapy mediated by L1CAM inactivation.

Recently, translational analysis of the PORTEC-3 trial reported that women with high-risk p53-abnormal ECs significantly benefited from the addition of chemotherapy to adjuvant radiotherapy, while those with high-risk NSMP tumors did not [9]. It is conceivable that this lack of significance in the NSMP subgroup is due to its molecular heterogeneity, and that a further subclassification by means of additional biomarkers could reveal subgroups of patients who may actually benefit from different treatments. In this scenario, we can hypothesize that L1CAM might help in detecting NSMP patients potentially responding to platinum-based chemotherapy plus radiotherapy and suggest integrating L1CAM evaluation, along with molecular classification, into future prospective clinical trials. Moreover, the characterization of EC cell lines based on molecular classification and additional biomarkers would provide in vitro models, mirroring different EC subtypes, that could be targeted with specific drugs.

Among all the other additional biomarkers analyzed, only PR expression was significantly associated with a favorable outcome in the NSMP category. This result is in agreement with Karnezis et al. [12], who observed PR positivity to be a marker of improved prognosis in NSMP ECs. We observed that, although there were significant differences in the expression of markers among the molecular groups, none of them showed a significant effect on prognosis in any category, probably reflecting their more homogeneous molecular and clinical features.

## 5. Conclusions

The results presented herein, even if limited by the relatively small study cohort, indicate a potential benefit of L1CAM expression evaluation in adding prognostic and predictive information on the heterogeneous high-risk NSMP EC subgroup. Those findings are worth further confirmation in wider cohorts where molecular classification, response to platinum-based chemotherapy and prognosis are available, before considering the incorporation of L1CAM evaluation as a risk stratification marker in an EC clinical setting.

## Figures and Tables

**Figure 1 cancers-14-05429-f001:**
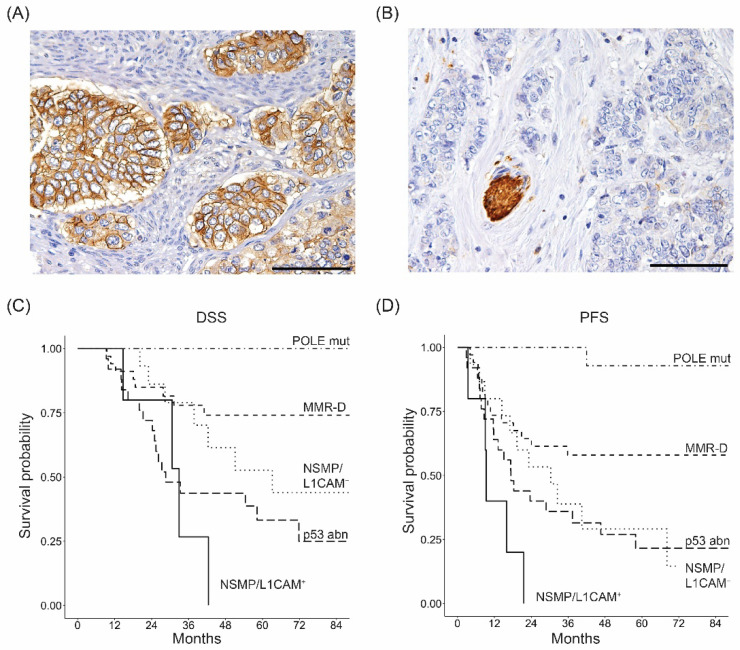
IHC staining for L1CAM in NSMP EC samples of the Brescia Cohort. Two representative cases of L1CAM-positive (**A**) and -negative (**B**) expression in tumor cells. The strong staining of the nerve (**B**) represents the internal positive control. Original magnification: 200×, scale bar 100 µm. Kaplan–Meier survival curves for disease specific survival (**C**) and progression-free survival (**D**) according to molecular classification, with the NSMP group stratified based on L1CAM positivity.

**Figure 2 cancers-14-05429-f002:**
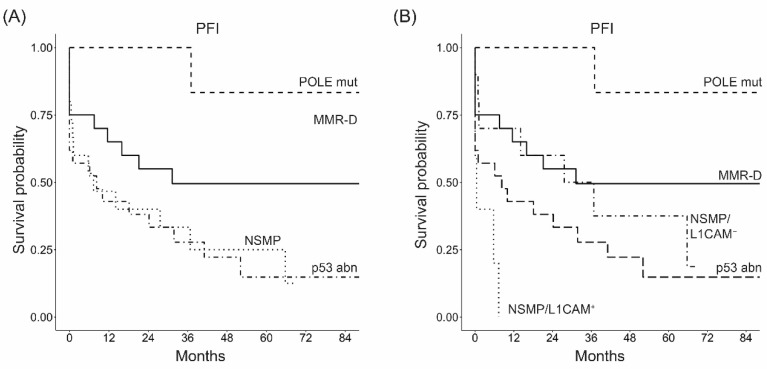
Kaplan–Meier survival curves for platinum free interval (PFI) considering molecular subgroups before (**A**) and after stratifying for L1CAM status within NSMP subgroup (**B**).

**Table 1 cancers-14-05429-t001:** Association between molecular subtypes and clinicopathological characteristics of the Brescia Cohort.

Clinical Annotations	POLE mut	MMR-D	NSMP	p53abn	Total	*p* Value
	n. 15 (16%)	n. 34 (36%)	n. 20 (21%)	n. 25 (27%)	n. 94 (100%)	
**Age (years)**						0.300 ^1^
Mean (SD)	62 (16)	63 (11)	66 (11)	68 (7)	65 (11)	
Median (Q1–Q3)	64 (54–74)	64 (55–71)	64 (60–75)	70 (64–72)	65 (59–73)	
**Histological type**						**<0.001 ^2^**
Endometrioid	12 (80%)	33 (97%)	17 (85%)	7 (29%)	69 (73%)	
Non-endometrioid	3 (20%)	1 (3%)	3 (15%)	17 (71%)	24 (27%)	
Unknown	0	0	0	1	1	
**FIGO Stage**						**0.015 ^2^**
I–II	11 (73%)	15 (44%)	4 (20%)	9 (36%)	39 (41%)	
III–IV	4 (27%)	19 (56%)	16 (80%)	16 (64%)	55 (59%)	
**Tumor Grade**						**0.024 ^2^**
G2	3 (20%)	5 (15%)	8 (40%)	1 (4%)	17 (18%)	
G3	12 (80%)	29 (85%)	12 (60%)	24 (96%)	77 (82%)	
**Myometrial invasion**						0.110 ^2^
M1 (<50%)	5 (33%)	6 (18%)	1 (5%)	7 (28%)	19 (20%)	
M2 (≥50%)	10 (67%)	28 (82%)	19 (95%)	18 (72%)	75 (80%)	
**Lymph node metastasis**						**0.006 ^2^**
Absent	10 (91%)	19 (63%)	4 (25%)	13 (65%)	46 (60%)	
Present	1 (9%)	11 (37%)	12 (75%)	7 (35%)	31 (40%)	
Unknown	4	4	4	5	17	
**LVSI**						0.110 ^2^
Absent	2 (13%)	1 (3%)	1 (5%)	5 (20%)	9 (10%)	
Present	13 (87%)	33 (97%)	19 (95%)	20 (80%)	85 (90%)	
**Risk Group**						**0.048 ^2^**
High–intermediate	9 (60%)	15 (44%)	4 (20%)	4 (16%)	32 (34%)	
High	4 (27%)	14 (41%)	13 (65%)	14 (56%)	45 (48%)	
Advanced Metastatic	2 (13%)	5 (15%)	3 (15%)	7 (28%)	17 (18%)	
**Adjuvant therapy**						0.051 ^2^
None	1 (7%)	0 (0%)	0 (0%)	0 (0%)	1 (1%)	
CT	5 (33%)	9 (26%)	6 (30%)	10 (40%)	30 (32%)	
RT	7 (47%)	14 (41%)	5 (25%)	3 (12%)	29 (31%)	
CT+ RT	2 (13%)	11 (32%)	9 (45%)	12 (48%)	34 (36%)	
**Time to recurrence after CT**						0.140 ^2^
Late-relapsing	6 (86%)	14 (70%)	7 (47%)	10 (45%)	37 (58%)	
Early-relapsing	1 (14%)	6 (30%)	8 (53%)	12 (55%)	27 (42%)	
Unknown	8	14	5	3	30	

^1^ Kruskal–Wallis rank-sum test; ^2^ Pearson’s Chi-squared test with simulated *p* value (based on 2000 replicates). Significant *p* values are in bold. First quartile (Q1); third quartile (Q3); standard deviation (SD); platinum-based adjuvant chemotherapy (CT); radiotherapy (RT).

**Table 2 cancers-14-05429-t002:** Univariable and multivariable survival analysis for Disease Specific Survival (DSS) and Progression Free Survival (PFS) according to molecular classification in the Brescia Cohort.

Variable	DSS				PFS			
	n. Events/n. Patients	HR	95%CI	*p* Value	n. Events/ n. Patients	HR	95%CI	*p* Value
**Univariable analysis**								
Molecular Groups								
MMR-D	8/34	1	-	-	14/33	1	-	-
POLE mut	0/15	0.10	0.00–0.78	**0.023**	1/15	0.11	0.01–0.83	**0.032**
NSMP	11/20	2.56	1.06–6.47	**0.037**	16/20	2.41	1.17–4.98	**0.017**
p53abn	17/25	3.62	1.63–8.70	**0.001**	19/25	2.3	1.15–4.61	**0.018**
**Multivariable analysis**								
Molecular Groups								
MMR-D	8/34	1	-	-	14/33	1	-	-
POLE mut	0/15	0.10	0.00–0.78	**0.024**	1/15	0.10	0.01–0.76	**0.026**
NSMP	11/20	1.98	0.79–5.14	0.142	16/20	2.43	1.13–5.22	**0.023**
p53abn	17/25	3.54	1.57–8.67	**0.002**	19/25	2.24	1.11–4.54	**0.025**
Age (year)	36/94	1.05	1.01–1.10	**0.013**	50/93	1.04	1.01–1.07	**0.010**
Grade (G3 vs. G2)	36/94	1.04	0.45–2.67	0.929	50/93	2.19	0.95–5.05	0.065
Stage (III-IV vs. I-II)	36/94	4.01	1.75–10.6	**<0.001**	50/93	3.76	1.86–7.58	**<0.001**

HR, Hazard Ratio; CI, Confidence Interval. Significant *p* values are in bold.

**Table 3 cancers-14-05429-t003:** Association between molecular subtypes and additional biomarkers of the Brescia Cohort.

Clinical Annotations	POLE mut n. 15 (16%)	MMR-D n. 34 (36%)	NSMP n. 20 (21%)	p53abn n. 25 (27%)	Total n. 94 (100%)	*p* Value
**L1CAM**						**<0.001 ^1^**
Negative (≤10%)	12 (80%)	31 (91%)	15 (75%)	8 (32%)	66 (70%)	
Positive (>10%)	3 (20%)	3 (9%)	5 (25%)	17 (68%)	28 (30%)	
**CTNNB1**						**0.003 ^1^**
Wild type	15 (100%)	28 (82%)	14 (70%)	25 (100%)	82 (87%)	
Mutated	0 (0%)	6 (18%)	6 (30%)	0 (0%)	12 (13%)	
**ER (% of positive cells)**						**0.023 ^2^**
Mean (SD)	37 (37)	51 (38)	35 (38)	18 (26)	37 (4)	
Median (Q1–Q3)	30 (5–70)	62 (6–84)	22 (0–80)	5 (0–25)	25 (0–75)	
**PR (% of positive cells)**						**0.002 ^2^**
Mean (SD)	24 (27)	36 (35)	33 (37)	6 (13)	3 (3)	
Median (Q1–Q3)	15 (5–35)	30 (3–69)	15 (0–65)	0 (0–5)	10 (0–40)	
**Ki67 (% of positive cells)**						**0.012 ^2^**
Mean (SD)	50 (20)	46 (22)	29 (22)	46 (26)	4 (2)	
Median (Q1–Q3)	55 (38–65)	42 (30–65)	20 (10–40)	38 (25–71)	40 (23–65)	
**ARID1A**						**0.008 ^1^**
Present	8 (53%)	19 (56%)	12 (60%)	23 (92%)	62 (66%)	
Loss	7 (47%)	15 (44%)	8 (40%)	2 (8%)	32 (34%)	
**Degree of inflammation** Absent/weak Moderate/prominent	2 (13%) 13 (87%)	19 (56%) 15 (44%)	15 (75%) 5 (25%)	16 (64%) 9 (36%)	52 (56%) 41 (44%)	**0.001 ^1^**

**^1^** Pearson’s Chi-squared Test with simulated *p* value (based on 2000 replicates); ^2^ Kruskal–Wallis rank-sum test. Significant *p* values are in bold. First quartile (Q1); Third quartile (Q3); Standard Deviation (SD).

**Table 4 cancers-14-05429-t004:** Univariable survival analysis for disease-specific survival (DSS) and progression-free survival (PFS) accounting for additional molecular/pathological biomarkers on the whole Brescia cohort of 94 EC patients.

Variables	N. of Patients	DSS			PFS		
		HR	95%CI	*p* Value	HR	95%CI	*p* Value
**L1CAM**							
Negative (≤10%)	66	1	-	-	1	-	-
Positive (>10%)	28	2.689	1.391–5.200	**0.003**	1.672	0.955–2.929	0.072
**CTNNB1**							
Wild type	82	1	-	-	1	-	-
Mutated	12	1.023	0.361–2.899	0.966	1.950	0.909–4.183	0.086
**Estrogen receptor**							
% of positive cells	94	0.987	0.976–0.997	**0.015**	0.996	0.988–1.004	0.301
**Progesterone receptor**							
% of positive cells	94	0.977	0.960–0.993	**0.006**	0.995	0.985–1.005	0.298
**Ki67**							
% of positive cells	94	0.994	0.980–1.009	0.454	0.991	0.978–1.004	0.177
**ARID1A**							
Present	62	1	-	-	1	-	-
Loss	32	0.429	0.194–0.945	**0.036**	0.578	0.306–1.092	0.091
**Inflammation**							
Absent/weak	53	1	-	-	1	-	-
Moderate/prominent	41	0.429	0.213–0.862	**0.018**	0.342	0.186–0.631	**0.001**

HR, hazard ratio; CI, confidence interval; Significant *p* values are in bold.

**Table 5 cancers-14-05429-t005:** Univariable and multivariable survival analysis for disease-specific survival (DSS) and progression-free survival (PFS), considering the molecular classification with NSMP subgroup stratified based on L1CAM positivity on the Brescia Cohort.

	DSS				PFS			
Variables	n. Events/n. Patients	HR	95%CI	*p* Value	n. Events/n. Patients	HR	95%CI	*p* Value
**Univariable analysis**								
Molecular Groups								
MMR-D	8/34	1	-	-	14/33	1	-	-
POLE mut	0/15	0.10	0.00–0.77	**0.023**	1/15	0.11	0.01–0.84	**0.034**
NSMP/L1CAM −	7/15	2.04	0.74–5.52	0.163	11/15	2.01	0.91–4.45	0.085
NSMP/L1CAM +	4/5	5.76	1.63–17.9	**0.009**	5/5	5.57	1.94–16.0	**0.001**
p53abn	17/25	3.65	1.64–8.75	**0.001**	19/25	2.39	1.19–4.78	**0.014**
**Multivariable analysis**								
Molecular Groups								
MMR-D	8/34	1	-	-	14/33	1	-	-
POLE mut	0/15	0.10	0.00–0.81	**0.027**	1/15	0.10	0.01–0.76	**0.027**
NSMP/L1CAM −	7/15	1.59	0.55–4.53	0.385	11/15	2.09	0.89–4.93	0.092
NSMP/L1CAM +	4/5	3.48	0.97–11.1	**0.056**	5/5	3.42	1.17–9.99	**0.024**
p53abn	17/25	3.56	1.58–8.67	**0.002**	19/25	2.26	1.11–4.57	**0.024**
Age (year)	36/94	1.05	1.01–1.09	**0.018**	50/93	1.04	1.01–1.07	**0.011**
Grade (G3 vs. G2)	36/94	0.93	0.39–2.43	0.870	50/93	1.97	0.82–4.71	0.130
Stage (III-IV vs. I-II)	36/94	3.82	1.65–10.2	**0.001**	50/93	3.65	1.80–7.42	**<0.001**

HR, hazard ratio; CI, confidence interval. Significant *p* values are in bold.

## Data Availability

All data not included in this published article are available from the corresponding author upon reasonable request.

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
