# Peer review of "Integrated Biomarker Analysis Reveals L1CAM as a Potential Stratification Marker for No Specific Molecular Profile High-Risk Endometrial Carcinoma"

_cancers, 2022, doi:10.3390/cancers14215429_

Round 1
Reviewer 1 Report
Thank you for asking me to review this manuscript.
The recruitment period is very wide and spans over 15 years given that these were consecutive cases. Does this mean that the Institution manages less than 10 cases/year?
There is no mention of the ethical approval given for this study. Did patients consent?
What was the ethnic breakdown of the study population?
No G1 tumours were included?
Why was MLH1 and MSH2 not tested? It still advised in the ESGO guidelines especially when abnormal MSH6/PMS2 staining is identified. This could have resulted in a number of the NSMP group being reclassified in the D-MMR group. Also without performing MLH1 hypermethylation studies it is not possible to determine the impact this may have had on outcome.
The results are interesting but there were only 20 patients in the NSMP group and of these only 5 were L1CAM+ve. Half of the NSMP/L1CAM group died of their disease (7/15) and on multivariate analysis the significance of L1CAM was lost.
Since L1CAM+ve was significantly associated with non-endometrioid histology and higher stage/grade at diagnosis, as compared to L1CAM-ve tumours, isn’t it possible that they were relapsing early due to this rather than L1CAM? The NSMP group should have been taken separately from the other groups to look at the factors that were associated with early relapse and survival and whether L1CAM offered any additional prognostic information in addition. It may be that L1CAM is a surrogate marker for high stage/grade but it is not possible to draw that conclusion from the data presented.
What was the platinum free interval in the L1CAM cases?
On page 10 I do not agree with the use of ‘highly’ to describe the significance level
In conclusion, the results are of interest but the conclusions are too strong given the very small patient numbers and the significantly greater number of higher stage/grade/non-endometrioid cases in the L1CAM+ve group.
Author Response
We want to thank Referee#1 for his constructive and insightful advice. We have addressed all the points raised by the reviewer as summarised below.
1. The recruitment period is very wide and spans over 15 years given that these were consecutive cases. Does this mean that the Institution manages less than 10 cases/year?
Reply: About 40 cases/year are treated in our Division of Obstetrics and Gynecology and the majority of endometrial cancers has an endometrioid histotype, early stage and low grade, belonging to the low or intermediate risk classes. In our study, we focused only on patients belonging to the high-intermediate, high or advanced-metastatic risk classes, as described in the first paragraph of the Materials and Methods.
- There is no mention of the ethical approval given for this study. Did patients consent?
Reply: The study was approved by the Research Review Board -the Ethics Committee- of Brescia, Italy (study reference number: NP3784) and written informed consent was obtained from all patients enrolled. These statements are reported at the end of the manuscript before the references, according to the Instructions for authors.
- What was the ethnic breakdown of the study population?
Reply: The study population consisted of Caucasian patients: 89 Italian (95%) and 5 Eastern European women. This information has been added in Table S1 (Table A1 in the revised manuscript).
- No G1 tumours were included?
Reply: I confirm that there were no G1 ECs in the population included in this study.
- Why was MLH1 and MSH2 not tested? It still advised in the ESGO guidelines especially when abnormal MSH6/PMS2 staining is identified. This could have resulted in a number of the NSMP group being reclassified in the D-MMR group. Also without performing MLH1 hypermethylation studies it is not possible to determine the impact this may have had on outcome.
Reply: The aim of our study was to identify the four molecular classes (NSMP, MMR, p53 abnormal and POLE mutated) in high-risk endometrial carcinoma. We were not focused on elucidating which was the molecular alteration at the basis of mismatch repair deficiency (mutation versus promoter hypermethylation and which gene was mutate/hypermethylated). Since Mismatch repair Genes work in couples (MLH1 with PMS2 and MLH2 with MLH6), the following guidelines recommend to test only two markers (MSH-6, PMS-2) in order to identify Mismatch Repair Alteration:
ESGO/ESTRO/ESP guidelines for the management of patients with endometrial carcinoma (Concin N, et al. Int J Gynecol Cancer 2021, doi: 10.1136/ijgc-2020-002230):
"Molecular classification is recommended to be performed by the TCGA surrogate using the diagnostic algorithm provided by Vermij et al.24 This diagnostic algorithm requires testing of three immunohistochemical markers (p53, MSH-6, PMS-2) and somatic mutation analysis of POLE (exons 9, 11, 13, 14)"
International society of Gynecological Pathology "POLE, MMR, and MSI Testing in Endometrial Cancer: Proceedings of the ISGyP Companion Society Session at the USCAP 2020 Annual Meeting (Casey L, International Journal of Gynecological Pathology 2021, 40:5–16, doi: 10.1097/PGP.0000000000000710):
"The MMR proteins occur as heterodimers with MLH1 pairing with PMS2 and MSH2 with MSH6. While MLH1 and MSH2 can stabilize in the cell by forming heterodimers with other proteins if the usual partner is absent, PMS2 and MSH6 can only exist stably in the cell in the presence ofMLH1 and MSH2, respectively. This has 2 important consequences. The first is that MMRd results in 4 typical MMR IHC patterns: (1) loss of both MLH1 and PMS2; this occurs in germline and acquired MLH1 defects; (2) loss of both MSH2 and MSH6; this occurs in germline and acquired MSH2 defects; (3) isolated loss of MSH6; this occurs in germline and acquired MSH6 defects; and (4) isolated loss of PMS2; this occurs in germline and acquired PMS2 defects. The second consequence is that testing for just 2 proteins, PMS2 and MSH6, can be used to screen for MMRd with accuracies equivalent to testing for all 4 proteins"
- The results are interesting but there were only 20 patients in the NSMP group and of these only 5 were L1CAM+ve. Half of the NSMP/L1CAM group died of their disease (7/15) and on multivariate analysis the significance of L1CAM was lost. Since L1CAM+ve was significantly associated with non-endometrioid histology and higher stage/grade at diagnosis, as compared to L1CAM-ve tumours, isn’t it possible that they were relapsing early due to this rather than L1CAM? The NSMP group should have been taken separately from the other groups to look at the factors that were associated with early relapse and survival and whether L1CAM offered any additional prognostic information in addition. It may be that L1CAM is a surrogate marker for high stage/grade but it is not possible to draw that conclusion from the data presented
Reply: Fitting a model on such a small subset of data would not provide any advantage. We can still create a more general model, but then focusing on a specific subset. Moreover, reducing the sample size subsetting the data would not be an effective strategy, as it would further reduce power. We cannot fully disentangle the effect of L1CAM from stage and grade, but this is always the case in observational studies: covariates share some degree of correlation. Nevertheless, the multivariate models suggest that L1CAM, regardless of FIGO stage and tumor grade, can still add some independent prognostic information. This is particularly evident with PFS, where L1CAM maintains its prognostic significance in multivariate analyses.
- What was the platinum free interval in the L1CAM cases?
Reply: The influence of L1CAM positivity on PFI in the NSMP subgroup has been described in the section 3.5 of the Results. Considering the whole cohort of patients, L1CAM-positive cases showed a significantly shorter PFI than L1CAM-negative ones (HR=1.86, 95%CI=1.03-3.45, p=0.049).
- On page 10 I do not agree with the use of ‘highly’ to describe the significance level
Reply: The term "highly" has been removed, as required.
- In conclusion, the results are of interest but the conclusions are too strong given the very small patient numbers and the significantly greater number of higher stage/grade/non-endometrioid cases in the L1CAM+ve group.
Reply: Consistent with the reviewer's suggestion, we looked for an independent cohort of high-risk EC patients to validate our findings.While we were able to retrieve an independent cohort that reflected our inclusion criteria, its clinicopathological characteristics and prognosis showed a substantial different pattern with a sensible lower event rate (recurrence and death), though higher in NSMP L1CAM-positive (25%) versus L1CAM-negative (10%), consistent with our cohort. Despite these differences, we were able to confirm that the predictive model including L1CAM provides a better characterization of the NSMP prognosis, compared to the baseline model, with a higher event rate in L1CAM-positive versus L1CAM-negative.
We added the following sentence to the Result:
“We validate the survival model for PFS on an independent cohort (ENITEC Cohort) of 47 high-risk EC patients, harboring 1 POLE, 7 MMR-D, 22 NSMP, and 17 p53abn tumors. We can observe a substantially different survival rate in the validation cohort, especially in the NSMP subgroup where only 18% of patients relapsed or died of disease. Due to this different survival rate, we recalibrated the survival models developed on the Brescia cohort using logistic calibration. The predicted 36 months PFS rates using the model accounting for L1CAM were 0.63 and 0.66 for the NMSP L1CAM-positive and L1CAM-negative respectively, with an observed PFS rate of 0.75 and 0.90. The model not accounting for L1CAM (based only on molecular classification) reported a predicted survival of 0.66 for L1CAM-positive and 0.64 for L1CAM-negative. While the estimates are still underestimating the true survival rate, due to the substantial difference between the two cohorts, adding L1CAM has actually achieved a better ranking within the NSMP subgroups, suggesting a lower PFS, coherent with the observed, for the L1CAM-positive group. Moreover, the concordance indexes (C-indexes) among observed and predicted survival for the NSMP subgroup from the two models were 0.75 for the extended model with L1CAM versus 0.68 for the baseline model. Thus, our results indicate that L1CAM expression, when added to the baseline model, improves the performance in predicting recurrence. “
Reviewer 2 Report
I read with great interest the Manuscript titled “Integrated biomarker analysis reveals L1CAM as a potential stratification marker for no specific molecular profile high-risk endometrial carcinoma”, which falls within the aim of the Journal.
In my honest opinion, the topic is interesting enough to attract the readers’ attention. The methodology is accurate, and the data analysis supports conclusions. Nevertheless, authors should include in the discussion section a consideration about some others potential endometrial cancer biomarkers such us non coding RNA including the following recent paper in the references section: PMID: 34830129, PMID: 34442102, PMID: 34540651 . Manuscript should be further revised by a native English speaker to improve clarity and readability. I want to inform You that I make a plagiarism check routinely, and I can confirm that Yours is an original writing.
Author Response
We want to thank Referee#2 for his constructive and insightful advice. We have addressed all the points raised by the reviewer as summarised below.
1. Nevertheless, authors should include in the discussion section a consideration about some others potential endometrial cancer biomarkers such us non coding RNA including the following recent paper in the references section: PMID: 34830129, PMID: 34442102, PMID: 33808791 .
Reply: We thank the reviewer for suggesting this interesting review (PMID: 34830129), which were added to the discussion (ref 32).
2. Manuscript should be further revised by a native English speaker to improve clarity and readability.
Reply: We thank the reviewer for the suggestion. The manuscript has now been reviewed by a native English speaker.
3. I want to inform You that I make a plagiarism check routinely, and I can confirm that Yours is an original writing.
Reply: Thank you for the thorough review of our manuscript.
Reviewer 3 Report
Integrated biomarker analysis reveals L1CAM as a potential stratification marker for no specific molecular profile high-risk endometrial carcinoma
Antonella Ravaggi et. al. Peer Review, MDPI Cancers
Summary: Ravaggi et. al. present a retrospective analysis of endometrial cancer patients, indicating that using L1CAM as a putative marker further stratifies the subtypes of endometrial cancers and thereby better predicts outcomes for patients.
The study is designed and performed satisfactorily and written up succinctly. The findings from this study can help a cohort of patients with endometrial carcinoma and hence warrants publication.
A few minor suggestions:
There are a few random font sizes through the initial sections of the paper, which may be due to file conversions, uploads, etc.
Line 189: “....36 (38.3%) were dead of disease and ….” could possibly be reworded as ‘...succumbed to the disease….’
Line 355: “....might help in discriminating patients benefiting from ….” could possibly be reworded as ‘... help in identifying patients …’
The authors acknowledge that this study is limited by a rather small cohort of patients. In order to overcome this, at least partially, and impart more confidence in the results presented, the authors could endeavor into analyzing publicly available data such as the TCGA, etc. to see if the findings in this study hold true in a larger subset of patient samples.
Perhaps outside the scope of this study, but it would be very interesting to see the levels of various markers in endometrial cancer cell lines. In light of the findings from the patient cohort, discussing the utility (or not) of these cell lines in functional assays could be very useful to the community at large.
Author Response
We want to thank Referee#3 for his constructive and insightful advice. We have addressed all the points raised by the reviewer as summarised below.
- There are a few random font sizes through the initial sections of the paper, which may be due to file conversions, uploads, etc.
Reply: In the original word version of the manuscript these errors were not present, perhaps there was a problem in the conversion of the file made by MDPI. We hope it doesn't happen this time.
- Line 189: “....36 (38.3%) were dead of disease and ….” could possibly be reworded as ‘...succumbed to the disease….’
Reply: The sentence has been changed as suggested.
- Line 355: “....might help in discriminating patients benefiting from ….” could possibly be reworded as ‘... help in identifying patients …’
Reply: The sentence has been changed as suggested.
- The authors acknowledge that this study is limited by a rather small cohort of patients. In order to overcome this, at least partially, and impart more confidence in the results presented, the authors could endeavor into analyzing publicly available data such as the TCGA, etc. to see if the findings in this study hold true in a larger subset of patient samples.
Reply: In agreement with the reviewer's suggestion, we evaluated the TCGA data. There are 42 high-risk patients in the NSMP group, but only 3 of them succumbed to the disease. Such a low number of events limits the possibility to validate the prognostic significance of L1CAM expression in the NSMP group. To our knowledge, no other publicly available databases with molecular classification and L1CAM expression quantification exist.
- Perhaps outside the scope of this study, but it would be very interesting to see the levels of various markers in endometrial cancer cell lines. In light of the findings from the patient cohort, discussing the utility (or not) of these cell lines in functional assays could be very useful to the community at large.
Reply: We agree with the reviewer that the characterization of EC cell lines based on molecular classification would provide in vitro models mirroring the different EC subtypes that could be targeted with specific drugs. Indeed, we recently demonstrated that a p53-abnormal EC cell line was more sensitive to carboplatin treatment after L1CAM silencing (Romani C, et al. Int J Cancer 2022, 151, 637-648) . To this aim, we added the following sentence to the Discussion:
“Moreover, the characterization of EC cell lines based on molecular classification and additional biomarkers would provide in vitro models mirroring different EC subtypes that could be targeted with specific drugs.”
Reviewer 4 Report
The authors addressed a clinically very important question of how to better stratify patients who have been diagnosed with a “no specific molecular profile” (NSMP) high-risk endometrial carcinoma (EC). They identified the often studied (but clinically rather under-utilized) expression of L1CAM as an additional and significant negative prognostic and predictive biomarker in this heterogeneous subset of NSMP-EC. The work has many strengths, e.g. long-term follow-up and many interesting results presented carefully and logically. One of the few weaknesses is the relatively small study cohort (94 patients), a limitation correctly recognized by the authors themselves. I have no real concerns, just a few suggestions for minimal improvements to this excellent work. Table 1 and the detailed description of the methods (as contained in the Supplement) should be provided in the Appendix, which would facilitate readability. Similarly, there is enough space (and reader interest) to move at least 1-2 tables into the main text and another 2-3 in the Appendix (currently in the Supplement). With other words, I would suggest giving more exposure to Figure S3, Table S7, S8, S9, S11. My impression is that the authors took the slightly simplistic approach that only variables that were significant in the univariate analysis were included in the multivariate analysis. This is probably the reason for not including LVSI in the multivariate analysis. Some readers would probably miss this variable, especially in the context of the study by de Freitas et al. 2018. Therefore, the criteria for inclusion in the multivariate analysis could be better explained and justified in the methods section. The discussion is well written, yet I wish the author would better embed his findings in the rich research landscape on L1CAM and high-risk EC. I think, that the discussion would benefit from at least some relevant and insightful papers, like e.g.: Guo M et al. High L1CAM expression predicts poor prognosis of patients with endometrial cancer: A systematic review and meta-analysis. Medicine (Baltimore). 2021 Apr 2;100(13):e25330. PMID: 33787629; Kommoss F et al. L1CAM: amending the "low-risk" category in endometrial carcinoma. J Cancer Res Clin Oncol. 2017 Feb;143(2):255-262.PMID: 27695947. de Freitas D et al. L1 Cell Adhesion Molecule (L1CAM) expression in endometrioid endometrial carcinomas: A possible pre-operative surrogate of lymph vascular space invasion. PLoS One. 2018 Dec 17;13(12):e0209294. PMID: 30557309Author Response
We want to thank Referee#4 for their constructive and insightful advice. We have addressed all the points raised by the reviewer as summarised below.
- Table 1 and the detailed description of the methods (as contained in the Supplement) should be provided in the Appendix, which would facilitate readability.
Reply: The Table S1 has been moved to Appendix A (Table A1). The description of the methods has been detailed in Appendix B.
- Similarly, there is enough space (and reader interest) to move at least 1-2 tables into the main text and another 2-3 in the Appendix (currently in the Supplement). With other words, I would suggest giving more exposure to Figure S3, Table S7, S8, S9, S11.
Reply: Table S11 has been moved to the main text (Table 4). The other tables suggested by the reviewer (Tables S7, S8, and S9) have been moved to Appendix A (Tables A2, A3, and A4, respectively), while Figure S3 has been moved to Appendix C (Figure A1).
- My impression is that the authors took the slightly simplistic approach that only variables that were significant in the univariate analysis were included in the multivariate analysis. This is probably the reason for not including LVSI in the multivariate analysis. Some readers would probably miss this variable, especially in the context of the study by de Freitas et al. 2018. Therefore, the criteria for inclusion in the multivariate analysis could be better explained and justified in the methods section.
Reply: We agree with the reviewer both that non-significant variables in univariate should be evaluated in multivariate and on the prognostic value of LVSI. Nevertheless, the vast majority of patients were LVSI positive (85 out of 94), so there is little information content in such an unbalanced comparison. Consequently, we did not include LVSI in multivariate analysis. We added this explanation in the Results section:
“Multivariable analysis included age, stage and grade as clinical characteristics. As the vast majority of patients were LVSI positive (85 out of 94), it was not included in the multivariate model.”
- The discussion is well written, yet I wish the author would better embed his findings in the rich research landscape on L1CAM and high-risk EC. I think, that the discussion would benefit from at least some relevant and insightful papers, like e.g.: Guo M et al. High L1CAM expression predicts poor prognosis of patients with endometrial cancer: A systematic review and meta-analysis. Medicine (Baltimore). 2021 Apr 2;100(13):e25330. PMID: 33787629; Kommoss F et al. L1CAM: amending the "low-risk" category in endometrial carcinoma. J Cancer Res Clin Oncol. 2017 Feb;143(2):255-262.PMID: 27695947. de Freitas D et al. L1 Cell Adhesion Molecule (L1CAM) expression in endometrioid endometrial carcinomas: A possible pre-operative surrogate of lymph vascular space invasion. PLoS One. 2018 Dec 17;13(12):e0209294. PMID: 30557309
Reply: We thank the reviewer for suggesting those relevant papers, which were added to the introduction (ref 14 and 15) and to the discussion, modified as follows:
“In the search for additional biomarkers useful for further NSMP prognostic stratification [12-20, 32], we found that L1CAM expression was significantly associated with poor outcome, regardless of tumor grade and FIGO stage. More specifically, NSMP/L1CAM-positive patients showed an extremely poor prognosis and features typical of aggressive tumors, such as the absence of ER/PR expression and ARID1A positivity. By contrast, the NSMP/L1CAM-negative were mainly ER/PR positive, ARID1A negative, and showed an intermediate prognosis, likely attributable to a more favorable underlying tumor biology. Our results are in agreement with previous studies, which showed, in cohorts including mostly low-risk tumors, L1CAM expression as an independent poor prognostic factor in EC patients [14, 15] and confirmed its potential in risk classification of NSMP subgroup. [33]. Our data suggest that L1CAM expression might be considered an independent adverse prognostic factor also in high-risk NSMP ECs.”